# Towards Smart Gaming Olfactory Displays

**DOI:** 10.3390/s20041002

**Published:** 2020-02-13

**Authors:** Georgios Tsaramirsis, Michail Papoutsidakis, Morched Derbali, Fazal Qudus Khan, Fotis Michailidis

**Affiliations:** 1Department of Information Technology, King Abdulaziz University, Jeddah 21589, Saudi Arabia; mderbali@kau.edu.sa (M.D.); fqkhan@kau.edu.sa (F.Q.K.); 2Department of Industrial Design and Production Engineering, University of West Attica, 104 31 Athens, Greece; mipapou@uniwa.gr (M.P.); fotismichailidis93@gmail.com (F.M.); 3Analyse et Commande Des Systemes, ENIG Unite de recherché Modelisation, 6000 Gabes, Tunisia

**Keywords:** olfactory display, olfaction, scent delivery, gaming olfactory display, gaming hardware, gaming gadget

## Abstract

Olfaction can enhance the experience of music, films, computer games and virtual reality applications. However, this area is less explored than other areas such as computer graphics and audio. Most advanced olfactory displays are designed for a specific experiment, they are hard to modify and extend, expensive, and/or can deliver a very limited number of scents. Additionally, current-generation olfactory displays make no decisions on if and when a scent should be released. This paper proposes a low-cost, easy to build, powerful smart olfactory display, that can release up to 24 different aromas and allow control of the quantity of the released aroma. The display is capable of absorbing back the aroma, in an attempt to clean the air prior to releasing a new aroma. Additionally, the display includes a smart algorithm that will decide when to release certain aromas. The device controller application includes releasing scents based on a timer, text in English subtitles, or input from external software applications. This allows certain applications (such as games) to decide when to release a scent, making it ideal for gaming. The device also supports native connectivity with games developed using a game development asset, developed as part of this project. The project was evaluated by 15 subjects and it was proved to have high accuracy when the scents were released with 1.5 minutes’ delay from each other.

## 1. Introduction

Recent advances in multimedia and virtual reality allow the delivery of very high-quality images and audio to the audience, offering an emerging experience. This experience can be enhanced even more by the use of additional dimensions such as scents [1]. Scents can be either directly related to a virtual object e.g., the smell of a strawberry or indirect e.g., the smell of spring. Smells that are directly related to an object are called “olfactory icons” while the indirectly related scents are called “olfactory smicons” [2]. Both of them enhance the quality of films, gaming and virtual reality and offer an even more immersive experience [3]. Presenting scents to the user, during such experience is possible by the use of devices called olfactory displays. Olfactory displays are devices, controlled by a computer, that deliver odorants to the human olfactory organ [4]. There are many such devices available at experimental level and few commercial products. However, few of them are cheap, easy to customize and extend and at the same time capable of delivering a variety of scents with high accuracy. Most olfactory displays, release scents based on a pre-set timer without any logic or awareness of the virtual phenomenon that scents are trying to simulate [5,6]. Researchers in [6], proposed an approach where the scents were released based on audio-visual information (screenshots and sample audio) collected from games at run-time. However, their research focused on the software that selects which scent is to be released and not on the hardware implementation. Another common problem and one of the reasons why this technology is not widely used, is that most olfactory displays do not include a mechanism for cleaning the air from previously released scents. This can result in a negative effect for the end user [6].

The aim of this research is to produce a low cost, extendable prototype device, capable of accurately presenting a variety of aromas to the audience. Additionally, the display will be able to absorb previously released scents via inverse rotation of its fans. A smart algorithm will prevent the same aroma from being released within a short period of time, saving the perfume and creating a more pleasant atmosphere, without compromising the quality of the experience.

The proposed solution is an olfactory display that can be placed in the environment and will utilize cheap open hardware components for delivering various scents to the audience. The prototype solution was named n-dimension as it is an extendable solution that can add dimensions to music, films, games and virtual-reality applications. The system relies on four servo motors that can deliver eight scents and two fans for pushing the air out as well as absorbing the air and the scent back, thereby cleaning the environment from previous scents. The device is expendable and it allows the users to add eight more servos and 16 more aromas without any programming. The device is expected to be placed in close proximity to the users in order to be effective.

A software application for controlling the display and a game development asset were developed as part of this project. The application was developed using C# and allows aromas to be released based on a certain time, target texts in subtitles, or input from other applications. The application is compatible and can accept input from the application in [6], enabling it to be used for gaming. A game asset was developed for the purpose or enabling a native link between games using this asset and the display. The asset is using invisible game objects with colliders to know when the player is near a game object and release the corresponding scent.

The next section (Background) presents the state of the art of olfactory displays. Section 3 presents the proposed device, the smart release algorithm, and the custom software. Section 4 evaluates the work. Finally, Section 5 concludes the work and provides for directions for future work in the area.

## 2. Background

Olfactory displays are divided into wearable and devices that are placed in the environment [7]. Wearable devices are those that the user wears like clothes, usually placed near the face. Wearable olfactory displays are divided into “On-body” and “Head-mounted”. An alternative approach is to use some external devices that are placed in the environment and they release or spray the user with some scent. Such devices include, “Natural vaporization”, “Air-canon” and “Fan/pump”. Figure 1, shows the categories of olfactory displays. This section provides a review of recent projects in each category.

On-body olfactory displays are systems that the user can wear, usually from the neck and below. Such olfactory displays can take the form of clothes, necklaces and backpacks that can release odorants. One recent project in this area is MicroFab’s “Pinoke” [8]. The “Pinoke” is the code-name of a prototype aroma generator in the form of a neckless, that can release fabricated odorants that result from the mix of four basic scents [7]. The main aim of this technology was to make gaming more realistic by enabling the players to smell the corresponding scene/virtual object of the game or virtual world.

Head-mounted systems depend on wearable devices such as masks, helmets that are placed on the head of the users and release odorants in close proximity from their nostrils. Researchers in [9], used an olfactory display connected to a virtual reality head mounted display in order to conduct an experiment where a user was eating a plain biscuit but because of the scents released from the olfactory system the user thought that he/she was eating biscuits with flavor. This system also used cameras and image recognition to know when the biscuit was in close distance as well as to know what scent it should spray.

Alternative to wearable technologies are devices that are placed in the environment. These technologies are categorized into Natural vaporization, Air-canons and Airflow-fans. Natural vaporization is the effect where the scent is released naturally without the use of any external mechanism such as heat. An example of this effect is the opening of the cover of a jar that will release the contained smell to the environment. It is worth noting that the scent must be strong or else the effect will be too weak and maybe it will be ignored by the subject. A problem with the approach is that the presentation is too slow [7] due to the slow speed of the air flows in a closed environment such as a room. An early example of natural vaporization was the Sensorama [1]. This device included four jars that contained aromas. A mechanical device could open the cover and allow the contained scent to be released.

Air-canons are named after the canon used to push out the scent, usually aiming the user. Air-canons can be implemented using various equipment such as fans, pump, blowers or other similar technologies [10]. The canons can be stationary and pointing at the user, or motorized with the ability to recognize the user and aim towards the nose. An example of this approach is an olfactory display developed in [11] that is using a camera and image recognition to identify the nose of the subject, move the canon to target it and push a smell towards it. Another example of this approach is presented in [10], where two canons were used to push air out with four different speeds towards each other forcing the scents to mix and then pushing the mix towards the user.

Air-flow solutions are dependent on fans, pumps, and blowers similar to the air-canons but they aim at the environment and not the user. Such devices can be placed in the roof of a room, on a desk or anywhere in the environment. An example of this approach presented in [12] was used for the development of a cooking game. The game enabled the players to cook using different ingredients such as butter, meat, onion, garlic, wine, curry roux, and spices into a virtual pan in a virtual reality environment. Once an ingredient was placed in the virtual pan, the system released the corresponding smell making the game more realistic [12]. Researchers in [13] developed an olfactory display that can be controlled by a tablet. The display uses heat and (a thermometer to control it) to release the perfume and a speed-controlled fan to push it to the environment. It can release up to eight aromas that the users can realise within seconds.

While these olfactory displays can enhance the user’s experience there is still a lot of room for improvements. Wearable technologies can serve only a single user and their presence is noticeable, hence it can disturb the user. The speed of the presentation is the same in all cases, although different users understand the scent after different times. That leads to low accuracy for some users. Another issue that lowers the accuracy was that after some time the scents were mixing as the previous scent was still there resulting in a maximum accuracy of 72% [8].

Another major category of olfactory displays consists of devices that are placed in the environment. One major advantage of such devices is that they may not be noticeable by the user; hence they tend not to disturb. However, this can also be a disadvantage as sometimes the scent may not reach the subjects or it so discreet it is not noticed. Another problem is the delay of the delivery of the scent to the nasal organ as well as the lack of functionality for cleaning an old smell before the new one is released.

Most of the applications from both wearables and “placed in environment” categories have been developed for specific experiments and it is not clear how they can be used for different applications. Most of them are expensive to develop or buy and are not extendable, so they are not ideal for simple experiments. A work presented by [2], attempted to address these issues and offer a simple, low-cost, easy-to-develop olfactory display that consisted of one Arduino Uno. The machine is capable of releasing one scent that was concentrated in a jar at a controlled time. They also used a fan for aiding the vaporization of the scent. This device however, is too simple to be useful as it can only deliver a single scent to a very short distance.

Some general problems with olfactory displays are that there are very few available perfumes hence it is not possible to simulate all the smells that the user can encounter in a virtual environment or even during a movie. Unlike colours that are produced by mixing basic RGB colours, there are no basic perfumes that can be mixed. This is possible for a very limited number of scents only. Another common problem is that, unlike image and sound, the smell cannot change as fast as these. For example, if a film is playing a scene in a forest, the olfactory display will spray the corresponding perfume. However, if the scene changes to an island in the middle of an ocean, even if the olfactory sprays the new perfume, the old one will still be present. Very little research has been done to correlate the scents with triggers events at the software level and especially with games and virtual environments [5].

In this research, we propose a simple but powerful olfactory display that is attempting to address or minimize the impact of the issues mentioned above and propose a novel solution for linking the scents with game scenes. Additionally, the proposed application offers a low-cost, extendable and easy-to-produce solution.

## 3. The Proposed System

### 3.1. System Architecture

The proposed solution includes hardware and software. The hardware includes: a control system, the actuators that are responsible for activating the spray, the jars with sprays that are responsible for releasing the aromas and two fans that help the scent reach the users as well as dispose of the old scent prior to the release of a new scent.

The software includes a Windows C# application that is used for controlling the time when the different scents should be released. Additionally, we developed a Unity3D [14] game development asset that activates corresponding scents when a user collides with a game object in the virtual world. The rest of this section describes in some detail the developed prototype.

### 3.2. The Hardware

Arduino Mega was used for controlling the hardware. The Mega has become a popular microcontroller for research purposes due to its low cost, open-hardware design and big community support with a lot of open source software. The board has 54 digital input/output pins (of which 14 can be used as PWM outputs), 16 analog inputs, 4 serial ports, a USB connection, a power jack and a reset button. We chose to use the Mega because of its inexpensive cost and this wide range of supporting features and above all, with the help of software support, it allows our device to be extended and support more aromas simply by adding more servos. In this project, the Arduino Mega is connected with four toy servo motors. Figure 2 and Figure 3 show a top down and a side view of the system.

The toy servo motors used in this project, are MG996R brushless DC motors that include decoder, gearbox and driver and are controlled directly from Arduino. The servo motors are responsible for pressing the sprays that deliver individual scents. The servos are installed between two jars so each servo can press two jars. Four scents can be released at the same time and eight scents can be released in total. The first servo controls the odorants of sea and mountain. The second servo controls the odorants of pine forest and mold. The third servo controls tobacco and orange. The fourth servo controls water (real water not just scent) and jasmine. It is worth noting that logically opposite aromas were controlled by the same servo. This is done because opposite aromas usually are not being released at the same time. However, if two aromas have to be released at the same time, the servo motor will press them with time difference of less than a second. The user can easily replace the aromas, simply by replacing the content of the jars. The Arduino is also connected with a L298N motor driver that is responsible for controlling two 12 V 140 mm × 25 mm fans, capable of outputting 77 CFM, 131 m^3^ speed of air each. The fans are responsible for aiding the sprays to push the scent to the environment as well as absorbing the scents back by rotating backwards. The speed of the fan is controlled by the software. The system is powered by a 450 W Corsair PC power supply unit, that provides 12 V to the motor driver and 5 V to the Arduino and the servos. The system connection schematic is presented in Figure 4.

As can be seen from Figure 4, the power supply unit directly provides 5 V to all servo motors, Arduino and 12 V to the L298n motor driver. The driver supplies the fans with 12 V. Arduino can control how many degrees the servo will rotate; hence how much scent will be released. It can also control the speed and direction of the fan. This way the fan can be used to either push the scent to the users or pull it back, cleaning in this way the air for the next scent. As can be seen from Figure 4, Arduino pins 42 to 50 are free and can be used by the user for adding additional servos. In the code these pins are attached to instances of the servo and the Arduino is programmed to control these pins like any servo pin. This allows the user to connect eight additional servos and 16 additional aromas without any programming making n-dimension truly expendable. All these functions are supported by the Windows C# application that was developed as part of this project.

## 4. Control Application

The Arduino controller can be controlled by a Windows application that has been developed as part of this project. Figure 5, is a screenshot of the n-dimension application.

At the beginning the user has to select a file that contains the effects to be played. The application can read files with “txt” extension that describe after how many seconds to play each effect. The first eight characters describe the time and are followed by a “*” symbol. The last part of the line is the aroma or the fan. The system has two fans, l for left and R for right. The number at the end describes the power of this command, e.g., the speed of the motor. If no number is present at the end, a light press or low fan speed will occur. The maximum number is four and it corresponds to maximum speed or a strong press of the sprays. If number five is provided the fan will operate with maximum speed towards the opposite direction and observe (clean) the air instead of blowing. The application uses a timer to know when to execute a command. Once a command is recognised, the application will send a corresponding code to the Arduino. The Arduino will then translate this code into the press of a spray with certain power or the rotation of a fan with a certain direction and speed. If a command is not recognised the application will print –1 and continue its operation. The user can pause or stop the application at any given time.

An alternative operation is the releasing of scents based on target words in the subtitle files. Figure 6 shows an overview of the process.

As can be seen in Figure 6, the application requires a list of target words and their synonyms. Each of these targets will correspond to the release of an aroma and are added manually by the user. Additionally, we have specified a list of what we defined as “pointer texts”. These strings help the system to determine if a scent should be released or not. If a sentence contains a pointer text and a target word or its synonym, then and only then will a scent be scheduled for release. This is done in order to avoid false releases. For example, consider the target word “forest” and the subtitle “I want to go to the forest”. The text contains the target word but not the pointer text, so no scent will be released. This is a desired behaviour, as the subject declare the desire to go to the forest but is not actually in a forest. It worth noting that the list of “pointer text” is filled by the developers and can be reused for all targets. However, the current list serves as a proof of concept and is not yet complete.

The olfactory display can accept input via the serial port of the Arduino from any application that follows the format specified above. This allows the application developed in [6] to control the olfactory display directly. This merge allows the display to release scents based on audio-visual content detected in the virtual world.

## 5. Reusable Game Development Asset

Unity3D is a very popular game engine with more than 1.3 million users [14]. The engine is used for developing games, virtual reality applications, mixed reality applications and simulations. Unity3D has built-in support for Oculus Rift and can produce applications that can work as both desktop and virtual reality at the same time. A Unity3D game development asset and a test scene was developed as part of this project. Figure 7, presents a screenshot from our test scene.

A prefabricated (prefab) asset, with an invisible game object and extended collider called “nDimension” was developed to allow the user to assign different aromas to the scene. As can be seen from Figure 7, the user can simply drug and drop the prefabricated asset to the scene, setup the colliders’ dimensions and write the name of the aroma that will be released every 60 s while the user is in the collider. All the collisions are automatically calculated by the engine so there is no need for the user to do anything else. Additionally, the system will try to determine the com port automatically and connect to the olfactory display. The system will get a list of all open com ports and attempt to connect to them one by one. If the connection is established, the system will send the character “n” and wait for reply. If the reply is “d” then this is the correct port, or else it will try the next one. Our scene also supports Oculus Rift virtual reality.

## 6. Smart Release Algorithm

Releasing a large quantity of aromas not only consumes the chemicals but also can create an uncomfortable environment [6]. The proposed approach supports inverse rotation of the fans for sucking back the aromas. We also recommend the use of non-persistence perfumes. However, after testing we discovered that it was not enough and there was a need to control the releases of the aromas. Also, it does not make sense to release an aroma again if its scent is still present. Finally, there may be cases of aromas that should not be released at the same time. To solve the above issues, we developed a smart release algorithm. The algorithm is hosted at the Arduino but at the beginning it requires some calibration values from the control application. These calibration values are the persistence times of each aroma, the list of aromas that should not be released together and the delay time before releasing a new aroma (if the delay is set to zero then the aromas will be released almost at the same time). This will enable the device to know how long it takes until the smell of each perfume is no longer detectable. These values are expected to be filled by the developers for all the supported aromas. This can be done through an empirical study, where the developers can release the aromas one by one and measure the time that it takes for the scent to disappear. The algorithm is presented in Figure 8.

The algorithm performs several checks before the release of any aroma. First it checks if the time elapsed since the release of the previous aroma is more than the delay value. If it is not, then it skips the release, or else proceeds to the next check. In the next check the system will check if the scent is still present due to an earlier release. This is checked by comparing how much has elapsed since the release of the same aroma with the aroma persistence time. If the elapsed time is less than the persistence time, then the aroma will not be released, or else the algorithm will go to the next check. In this check the system will check if there is a conflict between this aroma and previously released aromas. If there is no conflict, then the aroma will be released. In case of a conflict, the system will check how much time has passed since the release of the conflicted aroma in order to determine if its scent is still present. If the time passed is more than the persistence time of that aroma, this means that its scent is not present hence the new aroma can be released, or else the release will be skipped. This simple algorithm has proven to be very effective in terms of reducing the consumption of the perfumes as well as avoiding the creation of uncomfortable environments. The algorithm runs on the Arduino, so when active it will govern the release of all scents irrespective of the control application at software level.

## 7. Evaluation and Discussion

In order to evaluate the accuracy of our olfactory display we asked a game development company to conduct a “user experience” test. The company used our system and conducted an experiment with 15 participants (who were not involved with this project) to sit in close proximity (~1 m) to the device in a room of 30 square meters for 10 min and we released seven different scents with the same force. Each subject participated in the experiment at a different time. Table 1 shows when the different scents were released.

Table 1, consists of three columns. The first is the time that the aroma will be released. The second is the name of the aroma and the last is how strong or weak the aroma is. The subjects were provided with the list of aromas and they smelled each aroma prior to the experiment. However, they did not know in advance what time they would be released. The subjects were listening to music via headphones and were not looking at the device. Results of user experience evaluation demonstrate users were consistently able to identify “strong” scents. “Weak” scents were more difficult to identify unless a greater delay was applied. The results are summarized in Figure 9.

As can been seen in Figure 9, all subjects identified the scent of pine. Only two subjects managed to identify the ocean and after a few seconds they thought that we released the aroma of pine again. Twelve subjects identified the mountain. All of them identified the mold and the tobacco. Eleven people identified the orange and three people identified the jasmine, however three more people identified that we had released a different scent from the weak aromas but they were not sure what it was.

Based on these results, weak scents should be released about two minutes after a strong aroma, otherwise the user may not identify them. Strong aromas can be released with less delay, even after 30 s. If the weak aromas are released with less than 15 seconds’ difference, they will be mixed and the user will not be able to identify them. Strong aromas can be identified easier than weak ones. In general, weak aromas require about one and a half minutes to clear completely and strong aromas about three minutes.

All olfactory displays experience the difficulty of cleaning a scent when it is no longer required. Our approach used two fans to suck the scent back in order to solve this problem but some liquid from the perfume that fell in the environment was still generating a smell. However, applying the smart release algorithm had a much greater impact and almost solved the problem; however, it did not solve the problem of multiple aromas mixing, as the “non-conflicted” aromas were still released normally. Olfactory displays require aromas with low persistence time that can be disbanded faster in the environment.

The response time of the machine was less than half a second and the delivery to the user was very fast and satisfactory. The windows application that was used for the evaluation proved to be a useful tool and supports extra aromas with no coding. The new custom aromas that the user can add are supported by the windows application and can be called directly, making n-dimension easy to expand. The aromas schedule is controlled by a simple “txt” file that is very easy for anybody to produce. This txt file does not only describe what aromas should be released and when, but it also describes with what force each aroma will be released as well as how fast the rotation of the fan is and if it will blow wind or absorb the scents.

Additionally, the subtitle-based release mode was tested using a modified English version of the “.srt” file from an open source movie “Sintel” [15]. The modifications included adding the text of the appropriate effect at the end of the subtitle. The modifications and the test results are presented in Table 2. The system was tested with the smart release algorithm “On” and “Off”. As can be seen from Table 2, the device operated as expected.

In order to test the game asset we developed the Unity3D scene in Figure 7. In that scene we used one “nDimension” game object for the ocean. Every time the user was entering the area, the scent of the ocean was released. It proved easy to integrate n-dimension with Unity3D applications, enabling a more emergent user experience. The device tends to perform better in larger rooms (30 sqm+) than in smaller rooms, where some past released odors tend to persist. The cost of the device is low and it is easy to produce as it is based on open hardware equipment. The benefits of n-dimension are listed below.Smart release algorithm that governs the release of aromas, reducing their consumption rate and preventing over-releasing of aromas;Easy to expand;Easy to replace aromas;Limited ability to clear the air from previous aromas;Ability to control the quantity of the outputted perfume;Ability to control the speed of the generated wind and air flow;Able to serve a small group of users;Low cost (less than 150 USD);Windows application for advance time-based control or subtitles-based releases with no need for programming;Easy Unity3D integration.

The main limitation of this work is that currently there is a very limited list of “pointer texts” and aroma data, such as persistence times and conflict between aromas. The actual hardware is at an early experimental stage. More research is required in order to convert it to a working product.

## 8. Conclusions and Future Work

This paper presented a prototype olfactory display called n-dimension, capable of releasing eight aromas by default and 16 more as extensions. The machine is cheap to produce and to expand. The device is placed in the environment and can present the scents to a small group of users in close proximity. The device can control the quantity of the outputted air and scent and has limited capabilities of clearing the air from previous scents by sucking the air back. However, this was proven to have limited success. On the other hand, the introduction of a smart release algorithm, described in this paper practically solved the problem by preventing the over-release of scents. However, it did not solve the problem of multiple aromas mixing with each other. On the software side, the n-dimension is accompanied by a Windows application that determines when the various aromas will be released. Additionally, the application can release scents from information contained in subtitle files. A Unity3D prefab was also developed in this project, allowing easy integration between the hardware and Unity3D game engine. The device was evaluated for its accuracy of presenting different aromas by 15 subjects and it was found that it can deliver a clear scent as long as there is delay to the release of concurrent scents. The main limitations were that multiple aroma releases resulted in mixing of scents that were released within a small-time gap (when the “smart release” feature was not used). Another limitation is that currently it does not include enough information to release scents from any subtitle files and it does not have the persistence times and conflicts between various aromas.

In the future expansion of this work, we will try more aromas and complete the list of “pointing texts” enabling the scents to be automatically released using information from any subtitle file. We will also improve the quality of hardware and design 3D models so it can be reproduced easily by the community. The proposed solution is not the ultimate olfactory display, but it is a step forward and is proven to be a low-cost but powerful tool.

## Figures and Tables

**Figure 1 sensors-20-01002-f001:**
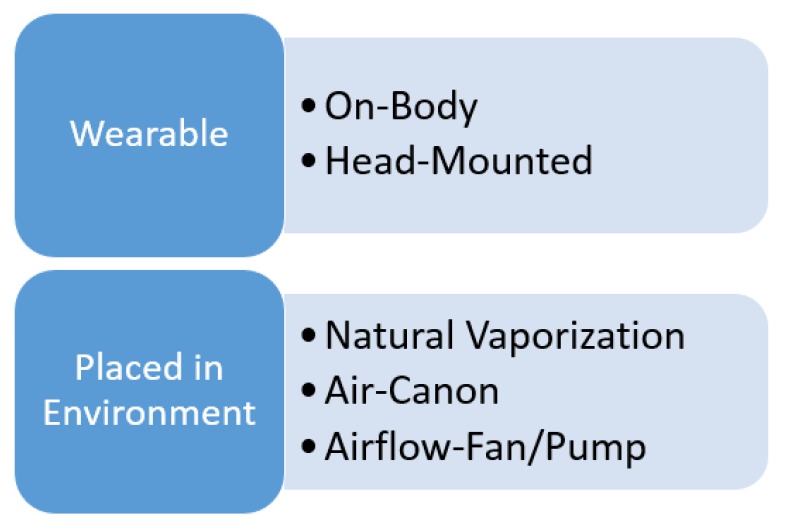
Categories of olfactory displays.

**Figure 2 sensors-20-01002-f002:**
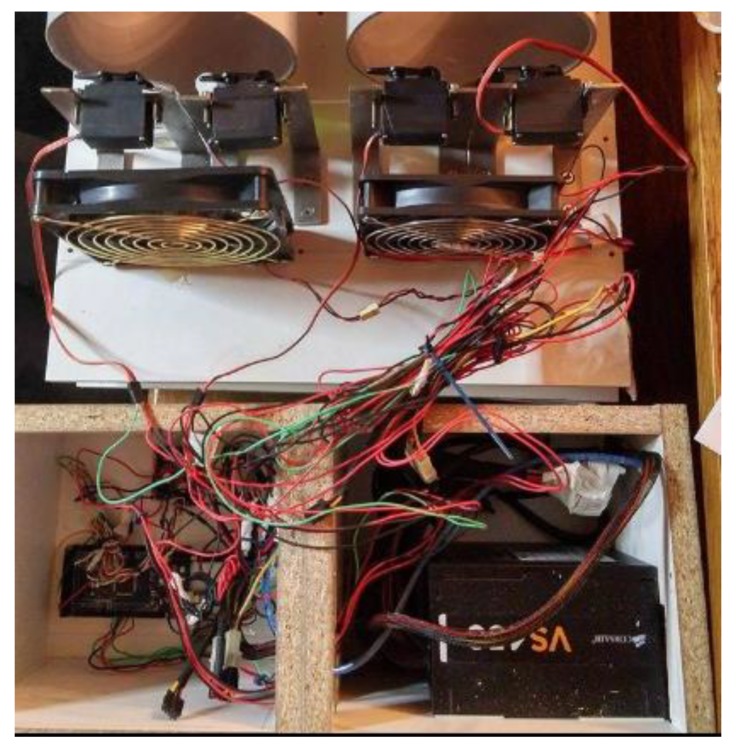
Top down view of the n-dimension prototype.

**Figure 3 sensors-20-01002-f003:**
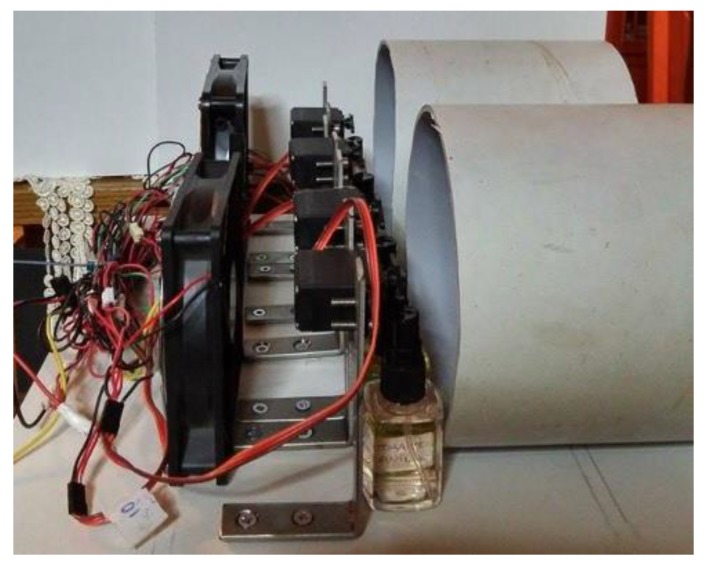
Side view of the n-dimension prototype.

**Figure 4 sensors-20-01002-f004:**
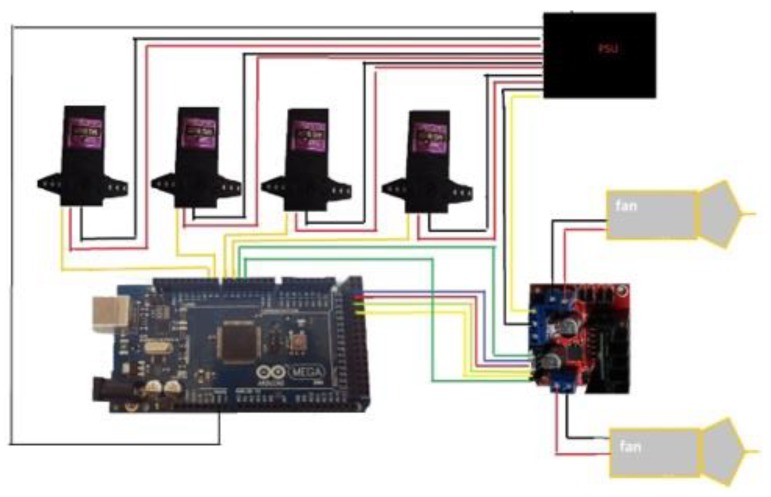
The project schematics.

**Figure 5 sensors-20-01002-f005:**
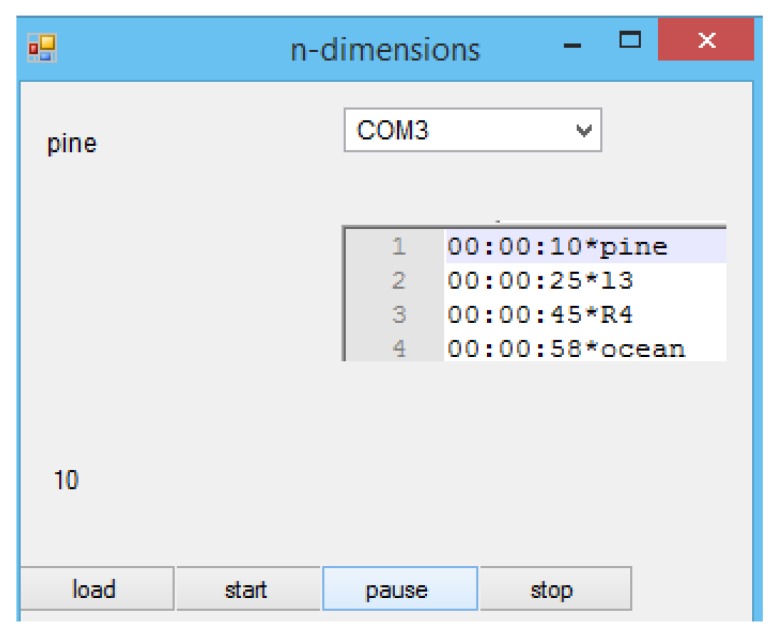
The olfactory control application.

**Figure 6 sensors-20-01002-f006:**
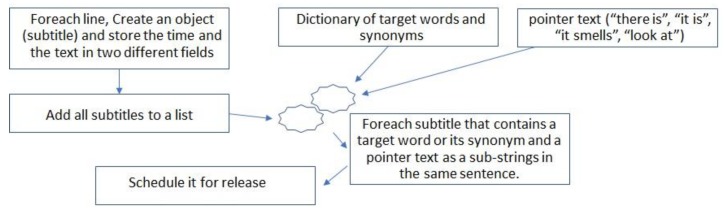
Releasing scents from subtitles.

**Figure 7 sensors-20-01002-f007:**
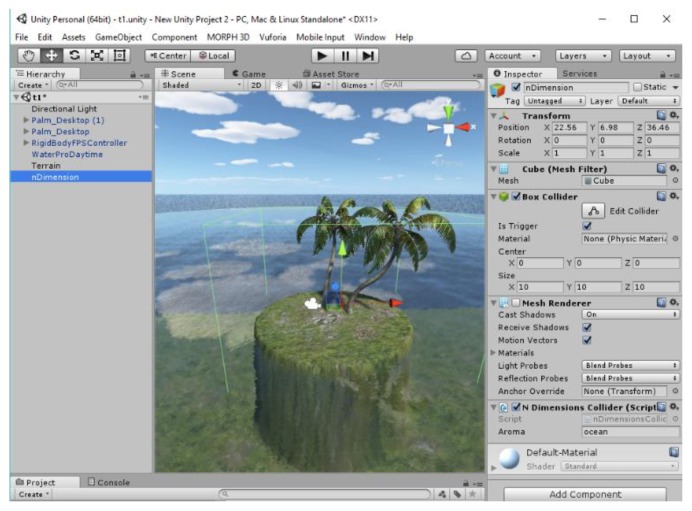
The Unity3D application.

**Figure 8 sensors-20-01002-f008:**
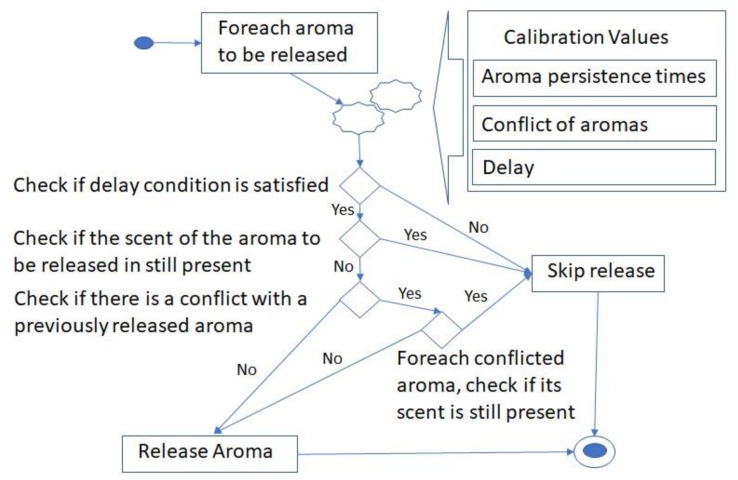
The smart release algorithm.

**Figure 9 sensors-20-01002-f009:**
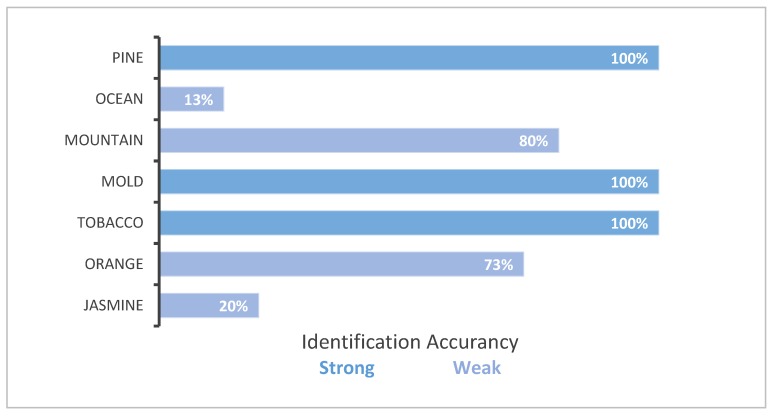
Percentage of users identifying the release of a scent.

**Table 1 sensors-20-01002-t001:** Aromas Schedule.

Time	Aroma	Strength
01:00	Pine	Strong
01:30	Ocean	Weak
03:00	Mountain	Weak
03:30	Mold	Strong
04:00	Tobacco	Strong
07:00	Orange	Weak
07:15	Jasmine	Weak

**Table 2 sensors-20-01002-t002:** Subtitle Mode Testing.

	No Smart Release		Smart Release	
Subtitle	Effect	Test	Effect	Test
00:00:18,250→00:00:42,500 (Strong Wind)	Fan (Max)	Passed	Fan (Max)	Passed
00:02:09,400→00:02:13,800 What brings you to the land of the gatekeepers (Smell of Tobacco)	Tobacco	Passed	Tobacco	Passed
00:02:15,000→00:02:17,500 I am searching for someone	No	Passed	No	Passed
00:00:18,000→00:02:22,200 Someone very dear? A kindred spirit? (Smell of Tobacco)	Tobacco	Passed	Tobacco	Passed
00:02:43,250→00:02:48,500 (Smell of Mold)	Mold	Passed	Mold	Passed
00:02:49,250→00:02:50,500 (Smell of Mold and Orange)	Mold and Orange	Passed	Mold and Orange	Passed
00:02:43,250→00:02:55,500 (Smell of Mold and Pine)	Mold and Pine	Passed	Mold and Pine	Passed
00:06:19,750→00:06:24,000 (Smell of Jasmine)	Jasmine	Passed	Jasmine	Passed
00:06:32,750→00:06:36,000 (Smell of Mountain and Strong wind)	Fan (max) & Mountain	Passed	Fan (max) & Mountain	Passed
00:07:25,850→00:07:27,500 I have failed (Smell of Tobacco)	Tobacco	Passed	Tobacco	Passed

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
