# Peer review of "Towards Smart Gaming Olfactory Displays"

_sensors, 2020, doi:10.3390/s20041002_

Round 1
Reviewer 1 Report
Authors make an adequate review of the state of the art about the prototypes developed in recent years to study the effects of olfactory stimulation.
One of the most outstanding features of this prototype seems to be that it includes a mechanism for cleaning the air from previously released scents, which presumably allows greater control of the variables. Another of its positive characteristics seems to be its low cost (although authors don't specify the difference in costs relative to the devices currently available in the market).
The main potential of this prototype seems to be the possibility of adapting the smell to the scene that the user visualizes, aspect still little developed in the available prototypes.
The contribution of this study is a breakthrough in research on the inclusion of olfactory stimulation in virtual environments and therefore I would recommend its publication; however, these systems for the stimulation of the sense of smell in virtual environments are still susceptible to improvements in order to emulate increasingly natural interactions.
Author Response
"Authors make an adequate review of the state of the art about the prototypes developed in recent years to study the effects of olfactory stimulation.
One of the most outstanding features of this prototype seems to be that it includes a mechanism for cleaning the air from previously released scents, which presumably allows greater control of the variables. Another of its positive characteristics seems to be its low cost (although authors don't specify the difference in costs relative to the devices currently available in the market).
The main potential of this prototype seems to be the possibility of adapting the smell to the scene that the user visualizes, aspect still little developed in the available prototypes.
The contribution of this study is a breakthrough in research on the inclusion of olfactory stimulation in virtual environments and therefore I would recommend its publication; however, these systems for the stimulation of the sense of smell in virtual environments are still susceptible to improvements in order to emulate increasingly natural interactions."
Thank you so much for your kind words and motivation. We are not aware of the cost of the other devices but we estimate that they cost more than our device. We also included the cost of our device (less than 150 USD).
Reviewer 2 Report
This paper describes a scent delivery system for enhancing media such as films, computer games, and virtual reality. The system prototype includes an algorithm for controlling scent released timing, to avoid release of conflicting scents and overuse of scents that are already active. The system also includes a fan system for aroma re-uptake. The system can release scents based on analysis of subtitles in film (using a reference library of trigger phrases), enabling the sensory enhancement of random input films without prior timing specifications. A Unity asset is also described to enable integration of scent triggers into the development of video games and virtual reality experiences.
Overall, the system and background is generally well-described and a range of applications are demonstrated in the paper, at least as a proof-of-concept. The main demonstration of the system with users seems to have been cut-off a bit, and the results of this evaluation could be presented better.
Major comments:
1. The start of Section 7 Evaluation and discussion seems to be cut off. It would be nice to understand the setup for this user experience evaluation a bit more. I’d recommend taking some time to introduce the experimental conditions before describing the results. Was the protocol for this study reviewed by any ethics board? Not sure if that’s really necessary, just curious. How were subjects recruited for the test? Were all the evaluations conducted in the same room? Approximately how big was the room? It is mentioned that the system was in “close proximity” to the subjects, but what is this distance approximately. Did the subjects complete the evaluation individually or in groups? Did they report the scents aloud or was this written onto a form?
2. Based on the setup for the user experience evaluation above, how big of a room would you think your system is most appropriate for? In other words, is the idea that this system could be good as an affordable solution for relatively small areas and small groups of people? It might be helpful to make the use case clearer to differentiate the system from the individualized/wearable systems and more expensive room setups.
3. As labeled in Table 1, how is the determination of “weak” vs “strong” scents made? Is this just a subjective classification, or is this related to the amount of scent that was being released by the system?
4. It would be nice to present the results of the user experience evaluation in a graph to see more clearly which stimulus presentations were accurately classified and which weren’t.
5. In the future, it would be interesting to vary the order of the scents being released.
6. In Table 2, we see the fans can be set to blow without a scent presentation (as triggered by the presence of “strong wind”). This is neat, but I don’t think this functionality was introduced earlier in the paper. It’d be nice to mention this earlier as a system feature.
7. While the background does a nice job of giving an overview of different types of systems similar to Murray et al. 2016), I feel like there are more recent systems, systems presented in the last five years, that could be mentioned. In general, there are not that many references in this paper.
Minor comments:
8. I’m sure the authors are aware of this and this would be corrected later, but I’ll just mention that the “Author contributions” is a template paragraph.
Author Response
Overall, the system and background is generally well-described and a range of applications are demonstrated in the paper, at least as a proof-of-concept. The main demonstration of the system with users seems to have been cut-off a bit, and the results of this evaluation could be presented better.
Major comments:
The start of Section 7 Evaluation and discussion seems to be cut off. It would be nice to understand the setup for this user experience evaluation a bit more. I’d recommend taking some time to introduce the experimental conditions before describing the results. Was the protocol for this study reviewed by any ethics board? Not sure if that’s really necessary, just curious. How were subjects recruited for the test?
An initial informal evaluation (not mentioned in the paper) was conducted by 15 students of gaming course of our department. However, in order to get a non-bias opinion we asked a game development company (infosuccess3d, Greece) to develop a mini scene using our asset and conduct a more formal evaluation using employees that were not involved with this project. Infosuccess3d, reported that they tested the system with 15 employees. Mr. Konstandinos Tsaramirsis (Company manager) monitor the experiments and record all the results. More details are now included in the paper.
Were all the evaluations conducted in the same room? Approximately how big was the room? It is mentioned that the system was in “close proximity” to the subjects, but what is this distance approximately. Did the subjects complete the evaluation individually or in groups? Did they report the scents aloud or was this written onto a form?
Thank you for your feedback. The paper has been updated to include all the answers.
Based on the setup for the user experience evaluation above, how big of a room would you think your system is most appropriate for? In other words, is the idea that this system could be good as an affordable solution for relatively small areas and small groups of people? It might be helpful to make the use case clearer to differentiate the system from the individualized/wearable systems and more expensive room setups.
Addressed “The device tends to perform better in larger rooms (30 sqm+) than in smaller rooms, where some past released odors tend to persist..”
As labeled in Table 1, how is the determination of “weak” vs “strong” scents made? Is this just a subjective classification, or is this related to the amount of scent that was being released by the system?
Strong smells = more persistent and easier to differentiate from other environmental smells. It was defined based on the perception of the developers.
It would be nice to present the results of the user experience evaluation in a graph to see more clearly which stimulus presentations were accurately classified and which weren’t.
Addressed
In the future, it would be interesting to vary the order of the scents being released.
Added to the future work section.
In Table 2, we see the fans can be set to blow without a scent presentation (as triggered by the presence of “strong wind”). This is neat, but I don’t think this functionality was introduced earlier in the paper. It’d be nice to mention this earlier as a system feature.
Thank you for your comment.
Addressed: “The fans are responsible for aiding the sprays to push the scent to the environment as well as absorbing the scents back by rotating backwards.”
While the background does a nice job of giving an overview of different types of systems similar to Murray et al. 2016), I feel like there are more recent systems, systems presented in the last five years, that could be mentioned. In general, there are not that many references in this paper.
We focused more on similar devices that can be placed in environment but it is always possible that we may have missed some references. Can you please provide us with any missing references you are aware of?
Minor comments:
I’m sure the authors are aware of this and this would be corrected later, but I’ll just mention that the “Author contributions” is a template paragraph.
Addressed
Round 2
Reviewer 2 Report
See attachment for review.

Author Response
First of all, I would like to take this opportunity to thank the reviewer for his comments.
All the concerns of the reviewer have now been addressed. In more detail:
In table 2, we change the word “Tabacco” to “Tobacco” Figure 10, has been updated